# Development and Testing of a Low-Cost Inactivation Buffer That Allows for Direct SARS-CoV-2 Detection in Saliva

**DOI:** 10.3390/vaccines10050730

**Published:** 2022-05-06

**Authors:** Brandon Bustos-Garcia, Sylvia Garza-Manero, Nallely Cano-Dominguez, Dulce Maria Lopez-Sanchez, Gonzalo Salgado-Montes de Oca, Alfonso Salgado-Aguayo, Felix Recillas-Targa, Santiago Avila-Rios, Victor Julian Valdes

**Affiliations:** 1Department of Cell Biology and Development, Institute of Cellular Physiology (IFC), National Autonomous University of Mexico (UNAM), Mexico City 04510, Mexico; brandonb@ifc.unam.mx (B.B.-G.); ncano@ifc.unam.mx (N.C.-D.); 2Department of Molecular Genetics, Institute of Cellular Physiology (IFC), National Autonomous University of Mexico (UNAM), Mexico City 04510, Mexico; sgarza@ifc.unam.mx (S.G.-M.); frecilla@ifc.unam.mx (F.R.-T.); 3Centre for Research in Infectious Diseases of the National Institute of Respiratory Diseases (CIENI/INER), Mexico City 14080, Mexico; dulce.lopez@cieni.org.mx (D.M.L.-S.); gonzalo.salgado.cieni@gmail.com (G.S.-M.d.O.); santiago.avila@cieni.org.mx (S.A.-R.); 4Laboratory of Research in Rheumatic Diseases, National Institute of Respiratory Diseases (INER), Mexico City 14080, Mexico; alfonso.salgado@iner.gob.mx

**Keywords:** SARS-CoV-2, COVID-19 testing, direct RT-qPCR, saliva

## Abstract

Massive testing is a cornerstone in efforts to effectively track infections and stop COVID-19 transmission, including places with good vaccination coverage. However, SARS-CoV-2 testing by RT-qPCR requires specialized personnel, protection equipment, commercial kits, and dedicated facilities, which represent significant challenges for massive testing in resource-limited settings. It is therefore important to develop testing protocols that are inexpensive, fast, and sufficiently sensitive. Here, we optimized the composition of a buffer (PKTP), containing a protease, a detergent, and an RNase inhibitor, which is compatible with the RT-qPCR chemistry, allowing for direct SARS-CoV-2 detection from saliva without extracting RNA. PKTP is compatible with heat inactivation, reducing the biohazard risk of handling samples. We assessed the PKTP buffer performance in comparison to the RNA-extraction-based protocol of the US Centers for Disease Control and Prevention in saliva samples from 70 COVID-19 patients finding a good sensitivity (85.7% for the N1 and 87.1% for the N2 target) and correlations (R = 0.77, p < 0.001 for N1, and R = 0.78, p < 0.001 for N2). We also propose an auto-collection protocol for saliva samples and a multiplex reaction to minimize the PCR reaction number per patient and further reduce costs and processing time of several samples, while maintaining diagnostic standards in favor of massive testing.

## 1. Introduction

Despite vaccination strategies against the Severe Acute Respiratory Syndrome Coronavirus 2 (SARS-CoV-2) successfully advancing in many countries, the appearance of hyper-mutated variants such as B.1.1.529 (omicron variant) [1], or lineages with higher viral titers such as B.1.617 (delta variant) [2], has severely impacted the progress made thus far to control SARS-CoV-2-associated disease (COVID-19), even on vaccinated individuals [3,4]. As COVID-19 confirmed cases continue to increase in countries facing recent waves of infection, massive testing campaigns constitute a key strategy to trace and stop the virus spreading, while vaccination continues. On the other hand, a possible outcome of the SARS-CoV-2 pandemic is that it will eventually become a seasonal health concern [5], which further highlights the relevance of developing safer, cheaper, faster, and more feasible universal testing strategies that may act as a surveillance approach in working environments and schools [6].

To this day, the gold standard for SARS-CoV-2 diagnosis is the detection of the viral genetic material by real-time RT-PCR, which involves the following: (1) collection of nasopharyngeal swabs (NPS) by specialized personnel; (2) purification of the viral genetic material with commercial kits; and (3) detection of at least two viral targets and a human target, usually RNAse *p* (hRP), by RT-qPCR into three separate reactions. These requirements bear several drawbacks if massive testing is expected, such as the following: (i) personal protective equipment (PPE) and specialized personnel are required to collect and extract RNA from the sample, (ii) commercial RNA extraction kits are expensive and their supply has been limited during the pandemic, and (iii) detecting targets into separate reactions results in the need of higher amounts of reagents and longer time to process several samples (and therefore have higher costs). All of these constitute limiting factors for massive testing.

The use of simpler-to-collect specimens for detecting SARS-CoV-2 supported by alternative processing procedures could aid in overcoming some of the limitations associated to NPS [7,8]. Raw saliva has been proven as a reliable source of viral genetic material as viral loads in this fluid can be equivalent to NPS [8,9,10,11]. Several processing sample alternatives have been reported for the use of saliva as specimen for SARS-CoV-2 infection diagnosis, including direct lysis by heating to avoid the RNA extraction step [12,13] or using commercial reagents [10], common buffers such as PBS or TE [13], Proteinase K [14,15], or guanidine hydrochloride [16] as lysis-inactivating solutions. However, some of these strategies have been reported to inhibit the RT-qPCR reaction to some degree [13,17], showing varying sensitivity for detecting viral targets in saliva. Thus, in most countries, RNA extraction-free protocols in saliva have not been considered to fully substitute COVID-19 diagnostics, and further studies are needed.

Different detergents have specific effects on cell components. For example, ionic detergents such as sodium dodecyl sulfate (SDS) and cetyltrimethylammonium bromide (CTAB) can solubilize lipid complexes and disaggregate or even denature protein complexes, while non-ionic detergents such as Tween or Triton are considered less harsh, as they can permeabilize the cell membrane, partially preserving the interaction of proteins with RNA and DNA [18,19,20]. The SARS-CoV-2 RNA genome is packed into 30–35 ribonucleoprotein complexes including the viral N protein [21,22] that may confer to the viral genome particular stability characteristics that could be relevant during sample processing. In this work, we systematically tested several buffer formulations and optimized Proteinase K and detergent concentrations to identify the conditions that, in conjunction to the addition of a broad-range RNAse inhibitor, improved viral genetic material detection in saliva by direct RT-qPCR without nucleic acid extraction. This optimized buffer (PKTP) allows for an easy sample inactivation protocol and circumvents the RNA extraction step, dramatically reducing processing time, costs, and biohazard risk. Additionally, the PKTP buffer can be used in several workflows, including standard RNA purification using commercial kits. We validated the use of PKTP buffer on 70 COVID-19 patients by comparing its performance vs. the CDC-validated RNA-extraction protocol on saliva samples. To facilitate the use of PKTP buffer, we propose a self-collecting sample strategy based on the use of graduated microcapillary tubes to pour saliva into PCR tubes containing PKTP buffer, which allows for direct heat inactivation of the sample, completely removing the need of manipulation of infectious specimens by laboratory personnel. To further reduce processing time and testing costs, we optimized a multiplex probe set to simultaneously detect two different viral genes and a human endogenous target in a single reaction.

## 2. Materials and Methods

### 2.1. COVID-19-Positive Saliva Samples

Saliva samples from COVID-19-positive adult patients were obtained at the National Institute of Respiratory Diseases (INER) in Mexico City, a tertiary referral hospital for COVID-19 during the pandemic. Participants’ ages ranged from 19 to 87 years old (average 40); they were 61% male, 39% female, and included health-care personnel and patients with different disease severity. For validation experiments, positive saliva samples for SARS-CoV-2 infection were selected according to viral Ct values previously assessed by qPCR at the moment of arrival to clinical care, using the CDC protocol [23]. We selected samples with Ct values between 15 and 45 to represent a wide range of viral loads. Saliva samples were collected by each participant by directly spitting into a sterile 1.5 mL tube. All samples were stored in 1.5 mL aliquots at −80 °C in an appropriate biosafety level 3 facility. Fresh aliquots were used for all validation experiments to minimize freeze-thawing cycles. All donors provided written informed consent for the use of remnant samples for research. The protocol for patient enrolment and sample donation was revised and approved by the Institutional Review Board of the INER (protocol B12-20).

### 2.2. PKTP Buffer

The PKTP buffer was composed of 0.1% Tween 80 (BP338-500 Fisher Scientific, Waltham, MA, USA), 5 mg/mL Poly Vinyl Sulfonic Acid sodium salt solution (278416 Millipore Sigma, St. Luis, MI, USA), and 200 µg/mL Proteinase K (Zymo Research, Irvin, CA, USA) in Ca- and Mg-free Dulbecco’s Phosphate-Buffered Saline buffer (DPBS L0615-500 Biowest, Riverside, MO, USA). The saliva samples were mixed with PKTP on a 1:1, 2:1, or 3:1 saliva:PKTP ratio and was heat inactivated (10 min at 96 °C) on a SimpliAmp Thermal cycler (Applied Biosystems, Waltham, MA, USA). Then, 4 µL of the inactivated saliva:PKTP mix was used for single- or multiplex RT-PCR reactions. The PKTP buffer was freshly prepared for all experiments except for the stability tests.

### 2.3. RT-qPCR Reaction

For single- and multiplex reactions, we used StarQ OneStep RT-qPCR kit (Genes2Life, Irapuato, Gto, Mexico) according to the manufacturer’s instructions. The final reaction volume was 20 µL, including 4 µL of StarQ Buffer 5X, 0.4 µL of StarQ Enzyme, 1 µL of probe mix containing 20X forward and reverse oligonucleotides, and 4 µL of heat-inactivated saliva in PKTP buffer (2:1). Nuclease-free water was used as non-template control (NTC). The oligonucleotides and probes were used at final concentrations indicated in Table 1, and the reactions were run in 384-well PCR plates (Bio-Rad, Hercules, CA, USA). RT-qPCR was performed on a BIO-RAD CFX384 Real-Time System under the following conditions: 50 °C for 15 min, 95 °C for 2 min, and 45 cycles of 95 °C for 15 s and then 60 °C for 30 s. For the CDC protocol, frozen saliva aliquots with a previous positive COVID-19 test result were thawed on ice, and 200 µL of the sample was extracted with QIAMP Viral RNA Kit, Cat. No. 52904 (Qiagen, Hilden, Germany) according to the manufacturer’s instructions. RNA was eluted in 50 µL of AVE buffer and 5 µL was used as template in a 20 µL reaction using the GoTaq Probe 1-Step RT-qPCR System Cat. No. A6120 (Promega, Madison, WI, USA). The N1-FAM, N2-FAM, and hRP-FAM oligo/probe assays were from the 2019-nCoV CDC EUA Kit, Cat. No.10006770 (IDT, Coralville, IA, USA). For comparison, PKTP reactions using the same samples of COVID-19 saliva were adjusted to 20 µL using 5 µL of inactivated saliva:PKTP mix (2:1). Reactions from CDC protocol and PKTP-inactivated samples were run together on Applied Biosystems™ 7500 Real-Time PCR System (ThermoFisher, Waltham, MA, USA). Other oligo/probe assays were acquired from T4Oligo (Irapuato, Gto, Mexico) with HPLC purification.

### 2.4. Self-Collection of Saliva Samples

We provided a kit for self-collecting saliva containing a calibrated microcapillary tube cat #2-000-050 (Drummond Scientific Company, Broomall, PA, USA), one 1.5 mL sterile microcentrifuge tube, and one PCR tube with 10 µL of PKTP. Saliva was collected by directly spitting into the 1.5 mL microtube, and 20 µL of the sample was transferred into the PCR tube containing PKTP by participants using the microcapillary tube. Participants were instructed not to drink, eat, smoke, or brush their teeth one hour before the test. The samples were heat inactivated (96 °C, 10 min) and stored for no more than 6 h at 4 °C.

### 2.5. Total Saliva RNA Integrity Test

Saliva samples were divided into two, and each aliquot was incubated in either PBS or PKTP buffer on a 2:1 saliva:buffer ratio for 12 h at 4 °C. After incubation, 200 µL of each sample was purified with QIAMP Viral RNA Kit, Cat. No. 52904 (Qiagene, Hilden, Germany) according to the manufacturer’s instructions and subjected to electrophoresis in a 2100 Bioanalyzer using an RNA Pico Kit, Cat. No. 5067-1513 (Agilent, Santa Clara, CA, USA).

### 2.6. Statistical Analysis

Variables in all data sets were tested for normal distribution (Shapiro-Wilk test) and homogeneity of variance (Levene test). Parametric tests (ANOVA, Pearson’s Product Moment Correlation) were performed as indicated, since data met the necessary assumptions. All statistical analyses were undertaken in GraphPad Prism 9.0.1 (1992–2012 GraphPad Software, Inc., San Diego, CA, USA) or the R statistical software version 3.6.3, using the PerformanceAnalytics package (V 2.0.4) for Pearson Correlation analysis [26]. Confidence intervals for the determination of the test sensitivity were calculated with MedCalc Software, using the “exact” Clopper–Pearson method [27].

## 3. Results

### 3.1. Detergent Screening for Direct SARS-CoV-2 Detection in Saliva by RT-qPCR

Viral titers in saliva have been reported to be equivalent or even higher than those in nasopharyngeal swabs from COVID-19 patients [9,10,11]. Therefore, we aimed to design a buffer formulation compatible with direct detection of SARS-CoV-2 by RT-qPCR in saliva avoiding RNA isolation, biohazard risks, and reducing processing times. Proteases have been used for lyse virus in saliva samples [15]; thus, our starting point was Proteinase K [800 µg/mL] in Ca^2+^- and Mg^2+^-free PBS (PK) supplemented with different detergents at 0.25% to improve buffer formulation.

SARS-CoV-2 virions are surrounded by a lipid bilayer that contains several highly glycosylated viral proteins (i.e., spike, envelope, and membrane) [28], while inside the capsid, the viral RNA genome is assembled as ribonucleoprotein complexes [21]. Therefore, we screened a broad range of detergents (ionic, non-ionic, mild-surfactants) to improve the release of viral and human genetic material without inhibiting the RT-PCR. We mixed SARS-CoV-2-positive saliva with PK buffer and different detergents prior to the heat inactivation of the sample. Then, 4 µL of the sample was directly placed in two individual RT-qPCR reactions to detect the SARS-CoV-2 E gene and the human RNase *p* gene (hRP), as an endogenous control (Berlin protocol, WHO probes). hRP detection was efficient in all formulations except for cetrimonium bromide (CTAB). N-Laurylsarcosine and Na-Deoxycholate showed the highest efficiency with a 100-fold increase vs. PK alone, followed by Tween 80 with a 10-fold change (Figure 1A, left). On the other hand, most detergents inhibited detection of the SARS-CoV-2 E gene except Tween 80, N-Laurylsarcosine, and SDS. It is noteworthy that Tween 80 improved SARS-CoV-2 E gene detection by 10-fold in comparison with PK alone (Figure 1A, right). Since PK + Tween 80 (henceforth referred to as PKT) was the sole combination that improved detection efficiency for both targets, we chose this combination for further optimization. 

To determine the precise detergent amount, we performed a concentration curve of Tween 80 in PK buffer. We mixed SARS-CoV-2-positive saliva samples with PK buffer containing different concentrations of Tween 80, heat inactivated the samples, and assessed each target individually. In the case of hRP, Tween 80 concentrations ranging from 0.15% to 0.35% resulted in a 30- to 40-fold increase in detection (Figure 1B, left), while SARS-CoV-2 E gene detection improved as Tween 80 concentration diminished, with a peak of a 40-fold increase at 0.1% (Figure 1B, right). Although the most efficient condition for hRP was 0.15% Tween 80, we selected 0.1% to favor the detection of viral genetic material since differences between 0.15% and 0.1% are not statistically significant for hRP.

### 3.2. Addition of an RNase Inhibitor Enhances Viral Genetic Material Detection in Saliva

Saliva is a source of RNases that may degrade the SARS-CoV-2 RNA, compromising its detection [29]. Thus, we added the thermostable RNase inhibitor Poly-Vinyl Sulfonic Acid (PVSA) to the PKT buffer (hereafter PKTP), a low-cost reagent that can enhance RNA preservation at low concentrations that has been previously used to lyse saliva and detect SARS-CoV-2 targets [30]. We performed a PVSA concentration curve and compared the detection efficiency between freshly inactivated samples and samples that were mixed with PKTP buffer and incubated for 3 h at room temperature before heat inactivation to test for RNA stability during benchwork time prior to sample processing. hRP detection was significantly increased in a PVSA dose-dependent manner, both with fresh and 3 h samples (Figure 1C, left). Similarly, SARS-CoV-2 E detection displayed a 10-fold increase in fresh lysates at 9 mg/mL of PVSA (Figure 1C, right), suggesting that higher concentrations of PVSA are advantageous. Next, to evaluate the stability of the RNA after inactivation, we repeated the PVSA concentration curve incubating the samples for 3 h after heat inactivation. This time, hRP detection was more efficient as PVSA concentration decreased, with a peak at 5 mg/mL (Figure 1D, left), while SARS-CoV-2 E was almost undetectable at higher concentrations of PVSA (Figure 1D, right). It has been suggested that PVSA may inhibit RT-qPCR at high concentrations, presumably as a consequence of its nucleic acid-mimicking characteristics [15,30]. However, since 5 mg/mL of PVSA displayed higher efficiency than PKT alone (Figure 1D, right), we concluded that this concentration of the RNase inhibitor is optimal for the detection of both the endogenous and the viral targets by RT-qPCR on saliva, protecting the viral RNA before and after heat inactivation of the sample.

To further assess the stability of RNA, saliva was mixed with PBS or PKTP (2:1) and incubated overnight at 4 °C, and then total RNA integrity was assessed using a Bioanalyzer instrument. The resulting electropherogram showed that the RNA in PKTP buffer was enriched in high molecular weight RNA molecules that are absent when saliva is incubated in PBS alone (Figure 1E), indicating that PKTP buffer is able to protect the RNA present in saliva for at least 12 h at 4 °C.

Finally, as proteinase enzymatic activity may be perturbed by the new buffer composition, we performed a titration curve of Proteinase K. Detection of hRP was significantly increased at lower concentrations of Proteinase K (Figure 1F, left). In contrast, that of SARS-CoV-2 E was clearly improved by Proteinase K addition at all concentrations, with the highest efficiency at 200 µg/mL (Figure 1F, right). Since this WHO-designed hRP primer set does not discriminate between gDNA and mRNA, we speculated that Proteinase K may particularly affect gDNA detection, while promoting a more efficient viral RNA release from viral particles contained in the saliva samples. As differences in hRP detection between 0 µg/mL and 200 µg/mL accounted for only a 0.5-fold decrease (about one single Ct unit), we set 200 µg/mL as the final concentration of Proteinase K to favor viral target detection.

### 3.3. Optimization of Saliva Lysis Conditions

Once we defined the optimal buffer formulation for the direct detection of a viral gene in saliva by direct RT-qPCR, we assessed mixing saliva with PKTP buffer at different ratios: 1:1, 2:1, and 3:1. Although the 1:1 ratio worked better for hRP detection, this condition displayed the greatest variation among replicates (Figure 2A, left). Otherwise, there were no significant differences in SARS-CoV-2 E gene detection among all tested ratios (Figure 2A, right). Hence, we chose the 2:1 saliva:buffer ratio as it was the most robust condition to lyse saliva samples in PKTP buffer.

Next, as Proteinase K displays its peak activity at 55–65 °C, we aimed to identify the best inactivating conditions by testing three different inactivating programs: (1) 10 min at 55 °C, (2) 10 min at 96 °C, and (3) 7 min at 55 °C followed by 10 min at 96 °C. hRP was clearly detectable with programs 2 and 3, although program 2 showed a significantly higher detection efficiency (Figure 2B, left). In contrast, SARS-CoV-2 E detection worked similarly between programs 2 and 3, with a slight improvement in program 3 (Figure 2B, right). The first program resulted in no detection of any target (Figure 2B), probably because non-inactivated Proteinase degrades the Taq polymerase and reverse transcriptase enzymes. Thus, we established 10 min at 96 °C as the simplest and standard working condition for inactivating saliva samples in PKTP buffer.

Additionally, we assessed the stability of the PKTP buffer by storing it at different temperatures. We stored aliquots from the same batch at room temperature (RT), 4 °C and −20 °C during 12 and 24 h. The stored buffer aliquots were used to inactivate SARS-CoV-2-positive saliva samples, and the RT-qPCR reactions were run in parallel vs. a fresh buffer batch. hRP detection resulted in a slight 0.2-fold decrease at 12 h in all temperatures, and approximately a 0.4-fold decrease (about one single Ct unit) at 24 h, when compared to the fresh buffer batch (Figure 2C, left). Of note, storage temperature differences did not reach statistical significance (*p* = 0.68 and 0.36, respectively). More variability was observed regarding SARS-CoV-2 E detection, since there was a non-significant 0.7-fold decrease when the buffer was stored at 4 °C (*p* = 0.086), while no change was observed when storing the buffer at −20° C both at 12 (*p* = 0.478) and 24 h (Figure 2C, right, *p* = 0.229). Notably, both hRP and SARS-CoV-2 E showed a statistically significant decrease at 24 h when compared to 12-hour-incubated samples at all temperatures (*p* = 0.0004 and 0.0347, respectively). This suggests that there is an advantage in preparing fresh buffer on the saliva sample collection day, rather than preparing it in advance and storing it. Altogether, these observations suggest that human and viral genetic materials display different stability profiles. We conclude that the PKTP buffer is stable at different temperatures facilitating different storage conditions and times.

Next, we evaluated the stability of the saliva:buffer mix before and after heat inactivation to define the suitability of the use of PKTP buffer in different workflows. First, we performed a stability assay of saliva:buffer mix for 3 and 6 hours at room temperature before heat inactivation. hRP detection showed no statistical differences among the different conditions, although there was an apparent 0.3-fold decrease (not significant, *p* = 0.24) at 6 hours (Figure 2D, left). In contrast, SARS-CoV-2 E detection displayed a trend to decrease over time, with a significant 0.6-fold decrease (*p* = 0.0097) at 6 hours (Figure 2D, right). Additionally, we checked the stability of saliva:buffer mix after heat inactivation for 12 and 24 hours stored at −20 °C. In these experiments, both human and viral targets showed a significant 0.5-fold decrease at 12 and 24 hours (Figure 2E, *p* = 0.001), which seems to be related to freezing-thawing cycles of the sample rather than the buffer [13]. In conclusion, the PKTP buffer can be introduced in different workflows, including storage before and after heat inactivation of samples, although a single Ct unit increase is to be expected when compared to freshly inactivated samples.

Finally, we tested sample processing with PKTP for both repeatability and reproducibility by testing intra-run replicates and intra-operator variability. We prepared aliquots from each of five SARS-CoV-2-positive saliva samples to ensure that two operators worked with the same samples and identical numbers of freezing-thawing cycles. Operators in two separate facilities performed three technical replicates (i.e., repeatability) for each PKTP-inactivated sample using the standard CDC-detection probes N1, N2, and hRP. Later, on a different day, each operator thawed samples right before the experiment and processed them using the PKTP buffer as described above (i.e., reproducibility). In all cases technical replicates yielded precise results (Figure 2F). Furthermore, there was no statistical difference in results for any of the three targets between operators (Figure 2F, *p* = 0.464), as determined by a two-way ANOVA test. Altogether, our results indicate that the PKTP buffer can be used by different technicians obtaining identical results, and that its use can be introduced into different work routines. 

### 3.4. Saliva Self-Collection Protocol

To further reduce the biohazard risk and to consider low-budget settings where no PPE nor trained personnel are available to collect samples, we developed a saliva self-collection strategy using the PKTP buffer. A commercially graduated microcapillary tube is used to place 20 µL of saliva into the tube containing PKTP buffer as represented in Figure 2G. The microcapillary tube has a mark at 20 µL (green line), and patients are instructed to directly pour saliva up to this mark. Then, the saliva contained in the microcapillary tube is placed into a regular PCR tube by the patient, containing 10 µL of PKTP buffer to guarantee the 2:1 saliva:buffer ratio. This mix is then incubated for 10 min at 96 °C in a thermal cycler for the heat inactivation of the samples, allowing for subsequent RT-qPCR without the need of pipetting, which significantly reduces the biohazard risk during sample processing. No differences in hRP values were observed by the use of microcapillary tube in comparison to pipetting saliva with a micropipette (Figure 2G, left).

### 3.5. Use of PKTP Buffer in a Clinical Setting

We evaluated the performance of our optimized PKTP protocol, which allows for the direct qPCR detection of SARS-CoV-2 in saliva without purifying RNA, in a hospital setting. This was assessed on 70 frozen saliva samples from previously diagnosed COVID-19 patients at the National Institute of Respiratory Diseases (INER), a tertiary referral hospital in Mexico City. We divided each stored sample into two: one half was heat inactivated in PKTP buffer and the other half was processed by the standard RNA-extraction-based protocol from the US Centers for Disease Control and Prevention (CDC). RT-qPCR reactions were run in parallel to detect viral N1 and N2, and human hRP targets were put into separate reactions using the CDC-designed probes (Table 1) to compare sensitivity between the two protocols. hRP was detected in all samples, showing a median increase of 3.5 Ct units when using the PKTP protocol (Figure 3A, top panel). Differences in the amount of template added to the reactions between protocols (6 times higher when RNA extraction is performed) may have accounted for the delay observed in the amplification plot. Similarly, Cts from viral genes from the purified RNA were lower than those from lysed saliva samples, possibly due to the aforementioned difference in template amount, with a mean difference of 3.64 Cts (N1) and 8.32 Cts (N2). Overall, both N1 and N2 viral targets were detected in approximately 80% of the PKTP-processed samples (Figure 3B,C, top panels): 60 and 61 samples were detected as SARS-CoV-2 positive with the N1 and N2 probes, respectively. This represents a diagnostic sensitivity of 85.71% (95% CI: 75.29–92.93%) for the N1 probe and 87.14% (95% CI: 76.99–93.95%) for the N2 probe using the PKTP protocol. 

We performed Pearson correlation analyses comparing the Cts of purified RNA vs. PKTP values. A good correlation was found with the three CDC probes (Figure 3A–C, bottom panels); Cts obtained using the hRP probe had a correlation coefficient of 0.82, while the correlation coefficients of Cts from N1 and N2 were 0.77 and 0.78, respectively. Our results show that saliva samples lysed with the PKTP buffer have a similar detection rate, although with a slightly lower performance in the RT-qPCR reactions, compared to the standard RNA purification-based protocol from saliva samples.

An increase in Cts when using the PKTP protocol could affect diagnostic sensitivity, especially when samples display low viral titers (>30 Cts); thus, we tested if our buffer could also be compatible with downstream RNA extraction using commercial kits. To address this, 10 COVID-19-positive saliva samples with varying initial Ct values were heat inactivated in PKTP buffer, their RNA was extracted with a CDC-approved kit [23], and the viral targets were assessed using the same CDC-designed probes (Table 1). All target genes were detected except for N2 in one low-titer sample (Figure 3D). This result suggests that the PKTP buffer can be used to inactivate saliva samples for further RNA extraction with commercial kits, pointing out the possibility to re-process suspicious samples when viral titers are too low to be detected by the direct PKTP-to-qPCR protocol or to use routinely the PKTP heat-inactivated saliva to extract RNA without biohazard risks.

### 3.6. Reducing the Number of RT-qPCR Reactions by Multiplexing

The standard CDC and WHO protocols use two different sets of probes for detecting SARS-CoV-2 genetic material. The CDC N1 and N2 probes target the nucleocapsid gene N, while the WHO probes test for two independent genes: the envelope gene E and the RNA-dependent RNA polymerase gene RdRp. In addition, both protocols target hRP as an endogenous control, requiring three independent RT-qPCR reactions per sample to emit a result. Hence, we aimed to reduce the number of reactions per sample to further reduce the use of reagents and to allow for more samples to be processed within the same RT-qPCR run, and therefore reduce processing time and costs, by using various probes with different fluorophores within a single multiplex reaction. To achieve this, we first evaluated the performance of the CDC and WHO probes labeled with common fluorophores for the detection of SARS-CoV-2 genetic material in serial dilutions of RNA extracted from saliva samples of three COVID-19 patients with different viral titers (Figure 4). We observed that the SARS-CoV-2 RdRp gene was consistently amplified with higher Cts than the N and E viral genes, which resulted in a lower sensitivity to detect low viral titers (Figure 4G). In contrast, the N1 and N2 probes showed the lowest Ct values in each condition (Figure 4A–C). The E gene probes showed similar kinetics to the N probes in most of the tested dilutions (Figure 4D–F). Thus, we selected N1 and E as viral targets to integrate the multiplex reaction in favor of testing two different viral genes. Amplification plots of these two viral targets showed consistency along serial dilutions of the RNA sample, while Cts exhibited linearity according to the viral titer (Figure 4). However, we observed that the sensitivity to detect the E gene at low viral titers was less compared to N1 and N2 (Figure 4).

Noteworthy, the E probe labeled with Cy5 detected low viral titers that were not detected when the probe was marked with other fluorophores (Figure 4D–F), raising the possibility that different fluorophore combinations could influence performance or detection during the RT-qPCR. To explore this, we designed two different multiplex reactions. In both, the highly sensitive CDC N1 probe labeled with FAM was included (reference probe). For multiplex 1, we used the E-Cy5 and hRP-HEX probes, and for multiplex 2, we used the E-HEX and hRP-Texas Red probes. We contrasted the performance of each probe from the multiplex 1 and 2 reactions with that of the corresponding singleplex reactions, using as templates the same serial dilutions of RNA extracted from saliva samples of three COVID-19 patients with different viral titers. There were no marked differences in the sensitivity, nor in the Cts, when the mentioned probes were used in singleplex or in multiplex reactions (Figure 5). In agreement with previous results, the E-Cy5 probe displayed higher sensitivity than the E-HEX probe in all conditions, providing similar Cts than the N1-FAM probe (Figure 4 and Figure 5A). However, both multiplex combinations generated the expected amplification plots and efficiently detected the three targets (Figure 5A).

Next, we evaluated the performance of multiplex 1 and 2 in a larger group of saliva samples from previously diagnosed COVID-19 patients enrolled at the INER Hospital in Mexico City and compared it with that of the original singleplex RNA-extraction-based CDC protocol. We used 26 stored samples, with a wide range of initial viral titers (Cts ranging from 15 to 35 for CDC N1 probe). Cts obtained with multiplex formulation 1 and 2 showed excellent correlations with Cts obtained from singleplex reactions (Figure 5B). Furthermore, all the viral probes showed high correlation between them, regardless of their use in singleplex or multiplex reactions (Figure 5B, Pearson correlation coefficients from 0.94 to 0.999). As expected, when reaction efficiencies are identical, there were no significant changes in Cts obtained between the three methods when using the RP probe (Figure 5C, *p* > 0.05) nor the N1 probe (Figure 5B,D, *p* > 0.05). These results suggest that viral gene detection is equally efficient using singleplex and multiplex tests. On the other hand, the E-Cy5 and E-HEX probes showed an excellent correlation (Figure 5B), although there is a mean difference of 1.8 Ct (E-Cy5 mean Ct = 26.6 (24.6–28.6), E-HEX mean Ct = 28.4 (26.6–30.2), *p* = 0.16, Figure 5E), which could possibly be due to differences in fluorescence detection efficiency of the two fluorophores in the system (different quantum efficiency in the detector, filter efficiency, etc.). Therefore, multiplex 1 was chosen for successive experiments.

Finally, we investigated whether the chosen multiplex 1 was compatible with the PKTP buffer by testing 20 frozen COVID-19-positive saliva samples that were previously heat inactivated in PKTP buffer. The viral targets N1-FAM and E-Cy5, as well as hRP-HEX, were evaluated by RT-qPCR in the same reaction (Figure 5F). All targeted genes were detected in the PKTP-inactivated samples. These results demonstrate that two viral genes and one endogenous control can be detected in a single RT-qPCR reaction using saliva samples without the need to isolate RNA, which reduces processing time, reagents, costs, and biohazard risks.

## 4. Discussion

Epidemiological surveillance of SARS-CoV-2 will be especially important, even in vaccinated populations, as recent evidence has revealed worldwide [3,4,31,32]. In this work, we optimized the composition of a buffer (PKTP) that is compatible with the RT-qPCR chemistry, allowing for direct SARS-CoV-2 testing in saliva without RNA extraction. This buffer is compatible with a heat-inactivation step that has been previously demonstrated to eliminate SARS-CoV-2 particles in saliva [13,33], minimizing the biohazard risks of handling the samples. However, RNA degradation is a concern when heating or storing samples, as saliva contains thermo-stable RNases that may compromise the diagnosis [29]. Our own data showed that storage of saliva resulted in low molecular weight RNA molecules indicative of degradation; thus, we included PVSA, a low-cost thermo-stable RNase inhibitor, in our buffer. Addition of PVSA resulted in an RNA size distribution enriched with larger fragments when storing saliva before heat inactivation and improved amplification of the E viral gene in a dose-dependent manner. However, after heat inactivation, high concentrations of PVSA seemed to inhibit viral target detection while cellular genes were less affected. Indeed, ribonucleoprotein complexing of the viral genome may account for this difference, but further studies, out of the scope of this work, would be needed to address this. Interestingly, our Proteinase K and detergent concentration screenings showed that cellular and viral genes have different behaviors: most detergents allowed for the amplification of hRP, but only Tween 80 at low concentration improved viral E gene amplification. An opposite trend was observed with Proteinase K, as a higher concentration decreased hRP detection, while the viral target was more resilient. These results suggest that viral RNA has particular characteristics that are not equivalent to the endogenous transcripts, which should be taken into consideration when developing novel testing strategies for coronaviruses. Overall, addition of PVSA, Tween 80, and Proteinase K at the optimized concentrations improved RT-qPCR detection of cellular and viral targets genes before and after inactivation of saliva.

It was reported that freezing saliva causes RNA degradation resulting in an increase of 2–3 Cts when detecting SARS-CoV-2 RNA by qPCR [13]. In our hands, the freezing effect on PCR performance of saliva mixed with PKTP was nearly a 0.5 Ct increase. Consistently, our stability tests showed that viral and cellular targets gained only ~0.5 Ct units when stored overnight at −20 °C after heat inactivation, or 3 h at RT before inactivation. Thus, our results indicate that samples could be stored/transported at a temperature range between RT and −20 °C without significantly affecting viral or human target Ct values. On the other hand, a stability test of the PKTP buffer itself showed that when frozen, storage does not compromise reaction performance. Therefore, PKTP use is suitable for different working routines without important performance differences to be expected. Additionally, a self-applicable saliva collection strategy using graduated microcapillary tubes could allow patients to directly deliver 20 µL of their own saliva into PCR tubes containing 10 µL PKTP buffer that can be directly heat inactivated in only 10 min with no further manipulation before qPCR. This significantly reduces the biohazard risk and the need of trained personnel, PPE, or dedicated facilities for RNA extraction.

We validated the use of the PKTP buffer in a clinical setting by paired comparisons between the CDC RNA-extraction-based protocol and our PKTP buffer method in saliva samples from a group of 70 COVID-19 patients. We showed that hRP was detected in all samples, while N1 and N2 were detected in approximately 80% of samples when using the PKTP protocol, representing a diagnostic sensitivity of 85.71% (CI 75.29–92.93%) for the N1 probe and 87.14% (CI 76.99–93.95%) for the N2 probe. The sensitivity of the PKTP protocol is higher than some rapid antigen test currently used in epidemiological surveillance in saliva (23% for Abbot Panbio [34], 44% for Biotime COVID-19 Antigen Test Cassette [35], or 77% for Fujirebio, Lumipulse G SARS-CoV-2 Ag [36]) and above the average (71.2%) of NPS rapid antigen test used worldwide [37]. 

Finally, to further simplify the protocol, increase the number of samples that can be processed in parallel, diminish the reagents required per sample, and reduce processing time and costs, we optimized multiplexed reactions to detect three targets in one RT-qPCR reaction. We showed no differences between using the viral and endogenous probes in singleplex reactions or multiplex settings. In addition, we showed that multiplexing is compatible with the use of the PKTP buffer without the need of RNA extraction. For the multiplex reaction, we chose viral genes N1 and E, as our data showed that these targets were more sensitive and reliable in comparison to RdRP target in samples with different viral titers. Although saliva is considered a poor specimen for influenza virus diagnosis, some have reported detection of influenza A and B in saliva [38]. Considering this, our method could be adapted to include other probe combinations or even to test for other pathogens that may be relevant in specific epidemiological contexts, such as SARS-CoV-2 and influenza virus co-infection, as previously reported [39].

## 5. Conclusions

Overall, our results showed that the protocol based on the PKTP buffer provides a good alternative to minimize the biohazard risk of handling potentially infectious samples, eliminating the need of trained personnel, PPE, or dedicated facilities to collect samples and to perform the RNA extraction step, although highly qualified personnel will still be needed to perform the PCR reaction and for data interpretation. It is important to take into consideration that we only implemented our saliva self-collection protocol in hospital and academic settings with saliva samples from adults. Implementation feasibility should be evaluated in different settings and age groups.

In our hands, false positive results were virtually nonexistent when assaying two viral targets, making the PKTP buffer an excellent alternative for massive testing in saliva not only in settings where trained personnel for sample collection are scarce and there are lacking biosafety facilities, but also where budget is a limitation and testing turnaround time is a major priority. The use of PKTP buffer could facilitate epidemiological surveillance in working environments, universities, or schools, even in children where NPS testing is not ideal.

The calculated cost of the PKTP buffer is less than one US dollar per sample, which in conjunction with multiplexed reactions represents a clear advantage in favor of massive SARS-CoV-2 qPCR testing using saliva samples. The trade-off between diagnostic sensitivity (especially for samples with low viral load, which have been suggested to represent low-to-zero infection risk [25]) and the overall costs and testing capacity should be considered in different contexts aiming to optimize the use of scarce human and material resources.

## Figures and Tables

**Figure 1 vaccines-10-00730-f001:**
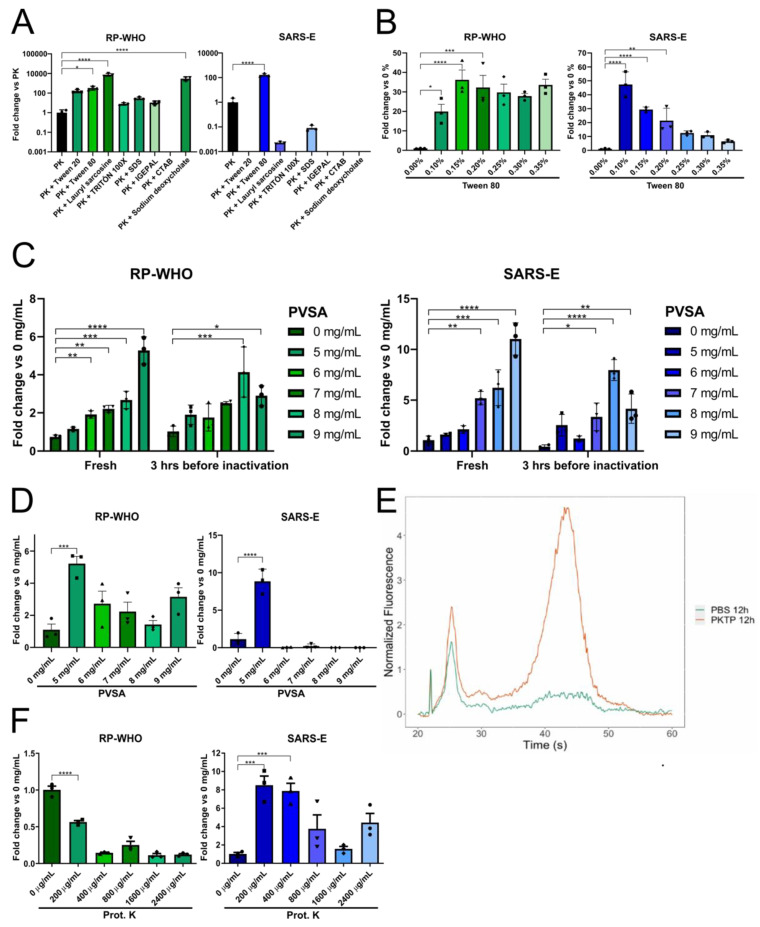
PKTP buffer allows for direct SARS-CoV-2 detection by RT-qPCR, protecting saliva from degradation. COVID-19-positive saliva was mixed in a 2:1 ratio with buffer and heat inactivated (10 min at 96 °C). Direct RT-qPCR detection of human RP (green) and viral E (blue) targets using WHO probes (see Table 1) was performed in 20 µL reactions. (**A**) Different detergents at 0.25% were screened for human and viral RNA detection. Fold change was normalized to the Ct values of PBS + Proteinase K (PK) only. (**B**) Detection efficiencies for both targets were assessed with different Tween 80 concentrations in PKT. Signal was normalized to PK with no detergent added. (**C**) Effect of PVSA concentration in freshly inactivated samples or samples incubated for 3 h at room temperature before heat inactivation. (**D**) Effect of PVSA concentration after heat inactivation. (**E**) RNA size distribution frequency in saliva samples incubated at 4 °C with PKTP or PBS alone assessed by a Bioanalyzer. (**F**) Proteinase K titration curve with full PKTP buffer formulation. Data were compared using a one-way ANOVA followed by Dunnett’s multiple comparisons test; (*, *p* = 0.05), (**, *p* = 0.01), (***, *p* = 0.001), (****, *p* = 0.0001). Bars represent means and 95% confidence intervals.

**Figure 2 vaccines-10-00730-f002:**
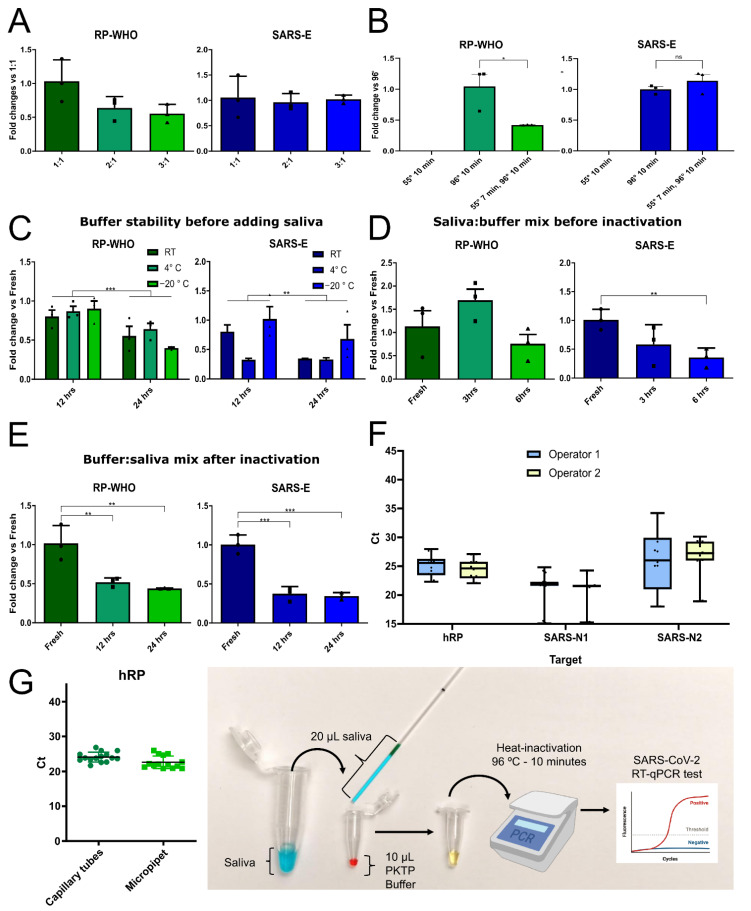
PKTP protocol optimization. Detection of hRP (green) and viral SARS-CoV-2 E (blue) targets using WHO probes (see Table 1) by RT-qPCR assessed on COVID-19-positive saliva. (**A**) Optimization of saliva:buffer mix ratio with PKTP buffer as lysing solution. (**B**) Optimization of the heat-inactivating program. (**C**) PKTP buffer stability determination by using buffer stored at different temperatures for 12 and 24 hours. (**D**) Stability tests of samples saliva:buffer mix 3 and 6 hours before PKTP heat inactivation stored at room temperature. (**E**) Stability test of samples stored at −20°C 12 and 24 hours after heat inactivation. (**F**) Repeatability and reproducibility tests of PKTP-protocol. (**G**) Proposed self-applicable saliva collection strategy using graduated microcapillary tubes to minimize biohazard risk. Left, comparison of hRP values obtained by pipetting saliva with a micropipette or capillary tube. Data were compared using a one-way ANOVA followed by Dunnett’s multiple comparisons test except for (**F**); (*, *p* = 0.05), (**, *p* = 0.01), (***, *p* = 0.001), (ns, not significant), Bars represent means and 95% confidence intervals. Part of the illustration was created with Biorender.

**Figure 3 vaccines-10-00730-f003:**
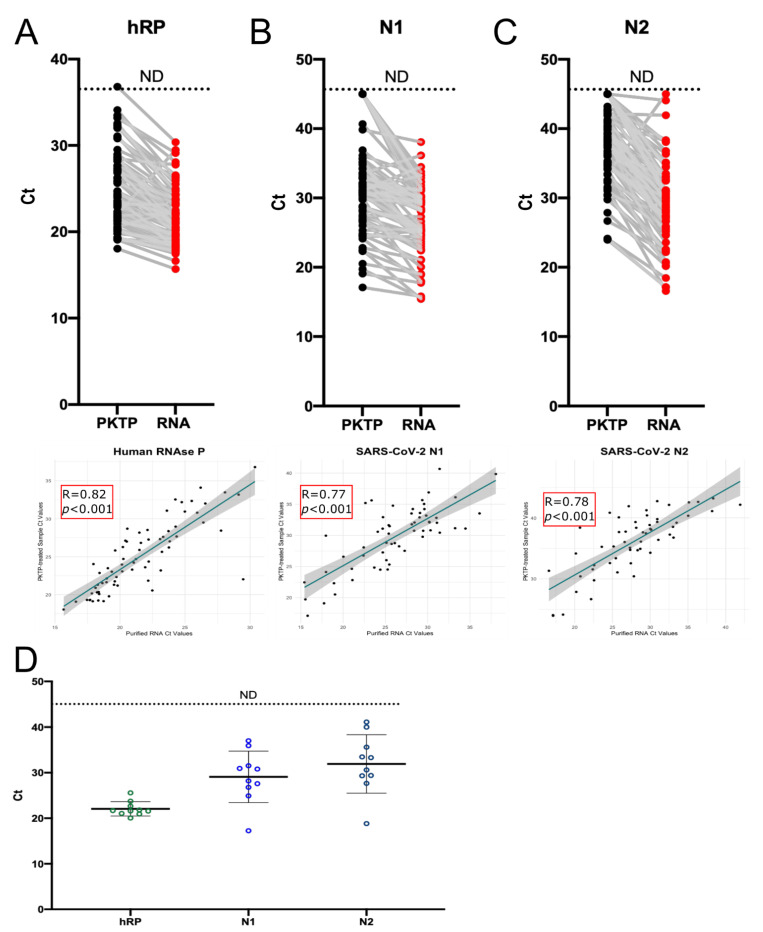
PKTP protocol performance in a clinical setting. Frozen saliva from 70 previously diagnosed COVID-19 patients at the INER hospital in Mexico City were used to compare human and viral RNA detection between the CDC RNA-extraction protocol (RNA) and our PKTP-inactivating protocol followed by direct RT-qPCR without RNA extraction. We used (**A**) hRP_FAM, (**B**) SARS-N1_FAM, and (**C**) SARS-N2_FAM (CDC probes, see Table 1) Cts-comparing protocols using paired samples (top panels). Lower panels display Pearson’s correlation test between protocols for each target. n = 70. (**D**) Ct values obtained by purifying RNA with a commercial kit (Qiagen) from PKTP-inactivated saliva samples.

**Figure 4 vaccines-10-00730-f004:**
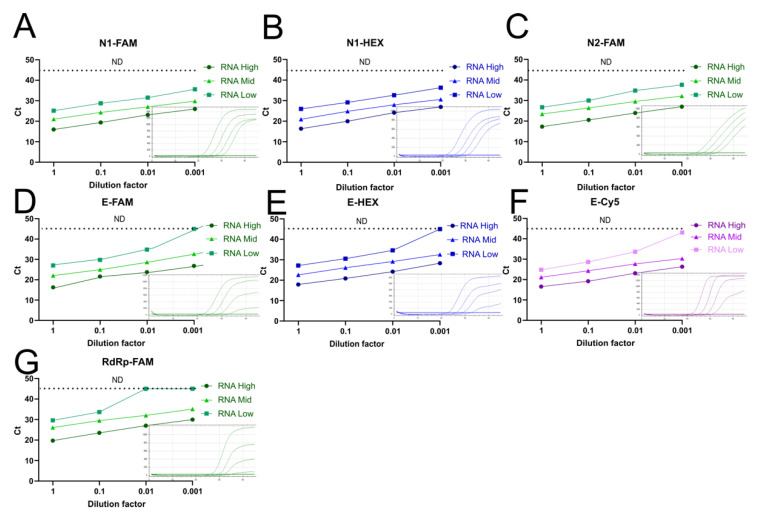
Detection efficiency of viral targets. RNA extracted from three SARS-CoV-2 saliva samples containing different viral titers (low, medium, high) was tested by RT-qPCR at serial dilutions. Ct values of each singleplex reaction using probes labeled with different fluorophores against viral targets (**A**,**B**) N1, (**C**) N2, (**D**–**F**) E and (**G**) RdRp (see Table 1 for sequences, fluorophores, and quenchers). Amplification plots of the serial dilutions corresponding to the RNA with a medium viral titer are included for each target gene.

**Figure 5 vaccines-10-00730-f005:**
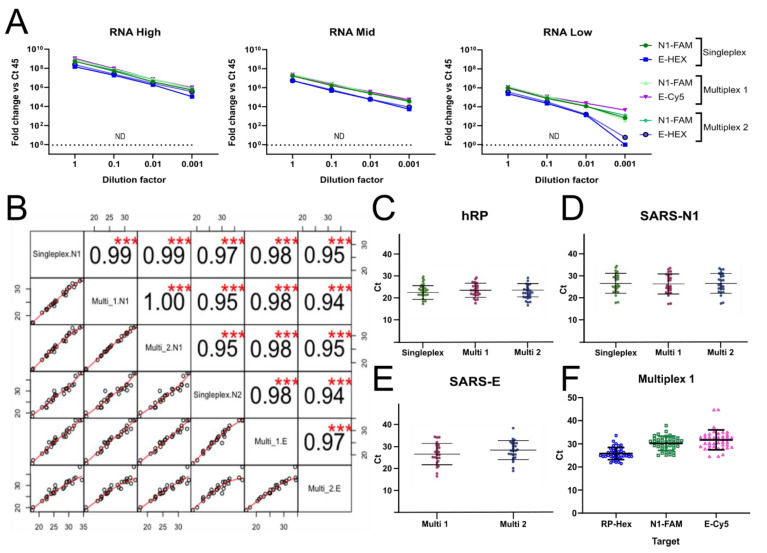
Multiplex reaction setting. (**A**) RNA from three different SARS-CoV-2 viral titer saliva samples (low, medium, high) were extracted and tested by RT-qPCR at serial dilutions. Graphs show the viral target detection efficiency in singleplex reactions in comparison to two multiplex reactions for three targets at a time. (**B**) Frozen saliva samples from 26 previously diagnosed COVID-19 patients at the INER hospital in Mexico City were used to compare the RT-qPCR signal between the CDC probes in singleplex and the multiplex reactions in the RNA-extraction protocol. Pearson correlation coefficients of Cts obtained for each target using singleplex or multiplex reactions (****, p* = 0.001). (**C**) hRP labeled with FAM (singleplex), HEX (Multiplex 1), or Texas Red (Multiplex 2); (**D**) SARS-N1 labeled with FAM in all cases (reference probe); and (**E**) SARS-E labeled with Cy5 (multiplex 1) or HEX (Multiplex 2) Ct values. (**F**) Multiplex-1 Ct values obtained for each target gene from 20 PKTP-inactivated positive samples, following the PKTP protocol previously described, without purifying RNA.

**Table 1 vaccines-10-00730-t001:** Oligonucleotides and probes used in this study.

Primer/Probe	Sequence 5′-3′	Concentration (nM)		Reference
		Singleplex	Multiplex	
WHO-hRP-F	AGATTTGGACCTGCGAGCG	500	200	[24]
WHO-hRP-R	GAGCGGCTGTCTCCACAAGT	500	200	[24]
WHO-hRP_FAM	[FAM]TTCTGACCTGAAGGCTCTGCGCG[BHQ-1]	125	-	[24]
WHO-hRP_TxR	[TxRd]TTCTGACCTGAAGGCTCTGCGCG	-	100	[24]
WHO-hRP_Probe P1_HEX	[HEX]TTCTGACCTGAAGGCTCTGCGCG[BHQ1]	-	100	[24]
2019-nCoV_N1-F	GACCCCAAAATCAGCGAAAT	500	400	[24]
2019-nCoV_N1-R	TCTGGTTACTGCCAGTTGAATCTG	500	400	[24]
2019-nCov_N1-P_FAM	[FAM]ACCCCGCATTACGTTTGGTGGACC[BHQ1]	125	100	[24]
2019-nCoV_N2-F	TTACAAACATTGGCCGCAAA	500	-	[24]
2019-nCoV_N2-R	GCGCGACATTCCGAAGAA	500	-	[24]
2019-nCov_N2-P_FAM	[FAM]ACAATTTGCCCCCAGCGCTTC AG[BHQ1]	125	-	[24]
WHO-SARS E-F1	ACAGGTACGTTAATAGTTAATAGCGT	400	400	[25]
WHO-SARS E-R1	ATATTGCAGCAGTACGCACACA	400	400	[25]
WHO-SARS E_HEX	[HEX]ACACTAGCCATCCTTACTGCGCTTCG[BHQ3]	200	200	[25]
WHO-SARS E_CY5	[Cyanine5]ACACTAGCCATCCTTACTGCGCTTCG[BHQ3]	200	100	[25]
WHO-SARS E_FAM	[FAM]ACACTAGCCATCCTTACTGCGCTTCG[BBQ]	200	-	[25]
RdRP_SARSr-F	GTGARATGGTCATGTGTGGCGG	600	-	[25]
RdRP_SARSr-R	CARATGTTAAASACACTATTAGCATA	800	-	[25]
RdRP_SARSr-P2	[FAM]CAGGTGGAACCTCATCAGGAGATGC[BBQ]	200	-	[25]

## Data Availability

All datasets used in this study are available from the corresponding author upon reasonable request.

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
