# Peer review of "Development and Testing of a Low-Cost Inactivation Buffer That Allows for Direct SARS-CoV-2 Detection in Saliva"

_vaccines, 2022, doi:10.3390/vaccines10050730_

Round 1
Reviewer 1 Report
Overall comment – the overall aim of this study as I understand is to, 1> use saliva directly as a sample, 2> reformulate a buffer that maintains sample integrity and obviates the need for RNA extraction, 3> optimize the RT-PCR to reduce the numbers of reactions & costs with this improved buffer? A paragraph or two in the Introduction as to why various detergents and other additives were selected would lay the ground for the reader as this is not clear in the Methods.
Introduction:
- There is more of a move towards decentralized and point-of-care testing versus centralized testing as the former offers faster turn-around-time versus continuing to centralize testing where there is an inherent delay in shipping samples to the testing lab?
- Various reagents that improve the detection of SARS-CoV-2 RNA are described in the Methods. A brief description & rationale for these choices in the Introduction would prepare the reader what the respective merits for your choices.
Methods:
- How many participants were selected? It would appear that from the participant descriptions no children samples were included? Can you comment on your rationale for excluding children, unless this approach is just for adults ?
- COVID positive samples – pg 2 line 90 -91, “samples positive were selected according to viral load”. Were all of these used in the subsequent optimization experiments of just a few and if so what was the criteria chosen as these samples must represent a range of Ct values ?
- Self-collection – pg 4, line 132 were all the saliva samples collected and stored at -20C for 6 hrs ?
- Self-collection – you conclude that pg 8 line 275, “PKTP buffer is best stored at -20” whereas for most participants do they have a -20 freezer to hand at home, otherwise the saliva samples would be of limited value ?
- Pg 5, line 169 – why select cetrimonium bromide ?
Results:
- Fig 1 A – What signal are you referring to - is it the Ct value, and if so how do you “normalize” it ?
- Fig 1D – It is very peculiar that PVSA had such a marked effect on the RNA of SARS-CoV-2 between 5 to 6 mg/mL concentrations whereas for the saliva RP gene target the effect was much less so – were these experiments run more than once for reproducibility ?
Discussion:
- Pg 16, lines 511-514, generally accepted that antigen assays for COVID & other respiratory agents display much poorer sensitivity than PCR assays, therefore this protocol should be compared to assays in its class, such as the Genxpert & ID Now
- Pg 16, line 503 -504 – I would disagree with such a broad statement, there is a still an issue of cross-contamination arising from the inherent sensitivity of PCR assays hence one still requires trained personnel that are familiar with this type of testing and as well the strict control during the pre-analytic stages prior & post to PCR testing
- Pg 16,line 525 – although co-infections of COVID & influenza have been reported, fair to say that saliva is a poor sample for influenza detection
Conclusion:
I return to two my earlier points 1> this buffer shows a reduction in effectiveness unless stored at -20 and most facilities schools, institutions, home self-collection, etc would generally have fridges but not -20C freezers available, and 2> the testing has to be done in a medium to large laboratory to accommodate the PCR technology, specialized personnel, quality assurance, etc so I fail to see the inherent advantages of the buffer as proposed by these authors unless it strictly relates to saliva testing without RNA extraction
Author Response
Please see the attachment
To REVIEWER 1:
We are grateful for the insightful comments from Reviewer 1, which have definitely improved our work.
Introduction:
- REVIEWER 1. There is more of a move towards decentralized and point-of-care testing versus centralized testing as the former offers faster turn-around-time versus continuing to centralize testing where there is an inherent delay in shipping samples to the testing lab?
REPLY: We thank Reviewer 1 for her/his comment, and we agree that decentralized testing has many advantages, however:
- Our protocol is not indented to substitute or compete with rapid antigen testing in decentralized point-of-care sites or home testing, but instead, to offer an improved sample processing strategy for qPCR testing that is still the most reliable SARS-CoV-2 testing strategy and is still massively performed worldwide in addition to rapid tests. Unfortunately, access to in-home testing is extremely limited in developing countries and most COVID-19 test are performed by PCR in certified laboratories and hospitals.
- The use of our PKTP-based self-collection protocol is intended to diminish: 1) the biohazard risks associated with sample collection in places where protective equipment and trained personnel for sample collection are limited, and 2) the time and cost-per-sample, by preventing the RNA extraction step and maximizing the number of samples that can be processed within a PCR run, considering that commercial reagents for NPS or saliva preparation and analysis are expensive and most of them have become scarce during the pandemic.
- We completely agree with Reviewer 1’s point noting that PCR testing still requires sophisticated equipment and has to be performed by trained personnel, but our PKTP-protocol is only intended to facilitate the sample collection and preparation (i. e. RNA extraction) stages. Regarding the delay in shipping samples, our results indicate that samples processed with PKTP are stable for 6 h at room temperature or up to 24 h when freezing is an option to transport samples (Figure 2) representing a clear advantage and supporting its potential use in different logistic scenarios.
- REVIEWER 1. Various reagents that improve the detection of SARS-CoV-2 RNA are described in the Methods. A brief description & rationale for these choices in the Introduction would prepare the reader what the respective merits for your choices.
REPLY: We thank Reviewer 1 for this important suggestion that improves the introduction of our work. Following his/her advice, we added a paragraph describing differences in the detergent’s behaviors in the Introduction (lines 70-78) and the rationale behind our choices are also explained along the Results section (lines 188-192). Adding this information definitely improved our work.
Methods:
- REVIEWER 1. How many participants were selected? It would appear that from the participant descriptions no children samples were included? Can you comment on your rationale for excluding children, unless this approach is just for adults?
REPLY: This is a valid observation raised by Reviewer 1 and we thank her/his comment. We worked with saliva samples donated by COVID-19 patients enrolled at the INER hospital in Mexico City that, at the moment of the study, were only adults within an age range between 19 and 87 years old and a sex distribution of 39% female, 61% male. This is now mentioned in the Methods sections (line 98-99) and a cautionary note at the Conclusion section (line 581-584)
- REVIEWER 1. COVID positive samples – pg 2 line 90 -91, “samples positive were selected according to viral load”. Were all of these used in the subsequent optimization experiments of just a few and if so what was the criteria chosen as these samples must represent a range of Ct values ?
REPLY: This is another valid observation of Reviewer 1 that we really appreciate. For validation experiments, positive saliva samples from INER Hospital COVID-19 patients were selected according to viral Ct values previously assessed by qPCR at the moment of arrival to clinical care. We then selected samples between 15 and 45 Ct, to represent a wide range of viral loads. This is now mentioned in line 100-102.
- REVIEWER 1. Self-collection – pg 4, line 132 were all the saliva samples collected and stored at -20C for 6 hrs ?
REPLY: We really thank Reviewer 1 for noticing this mistake. Saliva samples that were self-collected were not stored at -20 ºC but 4 ºC, for less than 6 hrs. This is now corrected in the manuscript in line 158. The only samples that were stored at -80 ºC were those collected at the INER Hospital where patients directly poured saliva into sterile 15 mL tubes.
- REVIEWER 1. Self-collection – you conclude that pg 8 line 275, “PKTP buffer is best stored at -20” whereas for most participants do they have a -20 freezer to hand at home, otherwise the saliva samples would be of limited value?
REPLY: We thank Reviewer 1 for this comment and the opportunity to clarify this point. We concluded that the optimal storing condition for the buffer is -20 ºC but this was for the buffer alone and not for inactivated saliva samples. On the other hand, our results on PKTP-inactivated saliva samples demonstrated that there are no statistical differences between 4º and -20º (Figure 2C), suggesting that even conventional freezers would be suitable for short-time storage of the samples. This was addressed in Discussion line 534. Additionally, and in accordance to Reviewer 1’s comment, we modified the mentioned statement in line 302-304. Additionally, to improve the clarity of the figure 2, we added titles to figure 2C (“Buffer stability before adding saliva”), 2D (Saliva buffer mix before inactivation”) and 2E (“Saliva buffer mix after inactivation”). We hope these changes improve the readers’ understanding of the figure.
- REVIEWER 1. Pg 5, line 169 – why select cetrimonium bromide ?
REPLY: We thank the Reviewer 1 for this comment, probably he/she noticed that CTAB was the only detergent to inhibit the detection of both, human and viral targets. We selected CTAB and all other detergents as all of these had been reported to efficiently release DNA from human cells and other microorganisms and with the objective to include a wide range of lysing conditions taking into consideration that while ionic detergents will disassemble and denature proteins, other detergents may preserve some ribonucleic complexes that may be beneficial for SARS-CoV-2 detection. In particular, CTAB has been used to release bacterial DNA from in saliva samples (PMID: 19069620), but as our data indicate, CTAB, -at least at the tested concentration-, inhibited the detection.
Results:
- REVIEWER 1. Fig 1 A – What signal are you referring to - is it the Ct value, and if so how do you “normalize” it?
REPLY: We are happy to clarify this. We refer to the corrected fold-change from raw Ct values that was normalized to the PK value as 1 to be taken as a basal detection without adding detergents. To improve the clarity of our manuscript, we now mention at the figure legend that the “Fold-change was normalized to the Ct values of PK” (Line 219)
- REVIEWER 1. Fig 1D – It is very peculiar that PVSA had such a marked effect on the RNA of SARS-CoV-2 between 5 to 6 mg/mL concentrations whereas for the saliva RP gene target the effect was much less so – were these experiments run more than once for reproducibility?
REPLY: This is another insightful observation of Reviewer 1 and we thank her/his comment. We were also very surprised to detect such different behaviors on viral vs. human target detection. This was a consistent observation. It has been suggested that PVSA may inhibit RT-qPCR at high concentrations presumably as a consequence of its nucleic acid mimicking characteristics, this was mentioned in line 248 and properly referenced in [15,30].
As Reviewer 1 noticed, it is striking that viral RNA detection is differently affected by PVSA concentration compared to the endogenous human gene target, however further experiments outside the scope of our work would be needed to understand these differences. We also suggest, that different stability profiles between viral and human genetic material (e.g. ribonucleoprotein complexing of the viral genome) may also account for this difference (Discussion line 522-524), but different experiments will be needed to address this. We hope that Reviewer 1 agrees with this.
Discussion:
- REVIEWER 1. Pg 16, lines 511-514, generally accepted that antigen assays for COVID & other respiratory agents display much poorer sensitivity than PCR assays, therefore this protocol should be compared to assays in its class, such as the Genxpert & ID Now
REPLY: We thank Reviewer 1 for her/his comments, and we completely agreed with this observation. Our work is aimed to aid PCR diagnostic and not to substitute or be compared to antigen assays. However, we only intended to point out that some antigen tests currently approved and used for epidemiological surveillance and diagnostics worldwide display varying sensitivity: as low as 23% for Abbott, Panbio in Spain (1), 44% for Biotime COVID-19 Antigen Test Cassette in Austria (2), 77.8% for Fujirebio, Lumipulse G SARS-CoV-2 Ag in Japan (3) or as high as 91% for the SalivaDirect initiative in Yale U (4). Moreover, systematic studies and meta-analyses of rapid antigen tests in NPS and Saliva showed varying sensitivity, (50-95%) (6) and an average of 71.2% (7), hence our protocol could be a valuable option since we validated a diagnostic sensitivity of 82.22% (95% CI: 67.95%-92%), not to mention that the cost-per-sample using our protocol is less than $1 USD and the processing time of the saliva samples is less than 10 minutes.
Therefore, and as we pointed out in the discussion, our sample collection strategy based on PKTP buffer is not intended to substitute gold-standard RNA-purification methods in places where this protocol is already established. Our proposal is that our PKTP-protocol can make a big difference in settings with budget restrictions or not equipped with biosafety facilities for sample processing, and our buffer is not intended to overcome any of the limitations arising from PCR inherent sensitivity or specific lab workflows.
(1) PMID: 33309541
(2) PMID: 33788280
(3) PMID: 33840598
(4) PMID: 33521748
(5) PMID: 34406838
(6) PMID: 33760236
(7) PMID: 34383750
- REVIEWER 1. Pg 16, line 503 -504 - I would disagree with such a broad statement, there is a still an issue of cross-contamination arising from the inherent sensitivity of PCR assays hence one still requires trained personnel that are familiar with this type of testing and as well the strict control during the pre-analytic stages prior & post to PCR testing
REPLY: We thank Reviewer 1 for her/his comment. We agree with Reviewer 1 that any PCR diagnostic protocol requires trained personnel and strict standards. Our PKTP protocol is not indented to modify the diagnostic requirement of the PCR but to aid in the initial purification steps and sample collection. We modified the original statements as Reviewer 1 suggested, and we mentioned that trained personnel and strict control is need for performing the PCR and result interpretation, clarifying that reduction in personnel only applies to sample collection. This was added in lines 580-584.
- REVIEWER 1. Pg 16, line 525 – although co-infections of COVID & influenza have been reported, fair to say that saliva is a poor sample for influenza detection.
REPLY: This is another valid concern of Reviewer 1 and she/he is right about that saliva has not been the primary substrate for testing influenzas virus. However, there is published evidence suggesting that in fact influenza can be detected in saliva samples (Reference #38 in the revised manuscript: Yoon, J et al. The use of saliva specimens for detection of influenza A and B viruses by rapid influenza diagnostic tests. J Virol Methods 2017, 243, 15-19, doi:10.1016/j.jviromet.2017.01.013), but of course, our PKTP-protocol would have to be validated in saliva of influenza patients. This is mentioned as a hypothetical possibility only at Discussion and clarified in lines 571-575.
Conclusion:
- REVIEWER 1. 1> this buffer shows a reduction in effectiveness unless stored at -20 and most facilities schools, institutions, home self-collection, etc would generally have fridges but not -20C freezers available,
REPLY: This is another valid observation of Reviewer 1 and we thank his/her comment. Although -20 ºC freezers are not universally available and it is true that the optimal workflow is to either, freshly prepare the buffer or store it at -20 ºC, our data also showed that the mixture of buffer and saliva, following the self-collection protocol, is stable for more than 3 hours before heat-inactivation (see figure 2D), providing enough time to collect samples at schools, institutions, etc. and to deliver them at the processing laboratory where qPCR is to be performed by trained personnel. To comply with Reviewer 1 comment, we added a cautionary note to the manuscript indicating that the optimal workflow is to freshly prepare the PKTP buffer (lines 301-303)
REVIEWER 1. 2> the testing has to be done in a medium to large laboratory to accommodate the PCR technology, specialized personnel, quality assurance, etc so I fail to see the inherent advantages of the buffer as proposed by these authors unless it strictly relates to saliva testing without RNA extraction.
REPLY: We thank Reviewer 1 for all her/his knowledgeable and insightful comments to our work. As we motioned throughout our manuscript and the previous comments of this letter, our PKTP-protocol is not indented to eliminate the need of highly-trained and capable personnel performing qPCR, but to: 1) eliminate the processing time of saliva-RNA extraction that in our experience is the most time consuming and “hands-on” part of the protocol, and 2) present an alternative for self-saliva collection using capillary tubes, once more reducing cost, time and the need of personnel for sample collection. In our opinion, eliminating the need of RNA extraction during the qPCR protocol is valuable in terms of time, cost and biosafety and was the ultimate objective of our work. Once more, we do not claim that our protocol would compete with rapid testing strategies, but offer an improved sample processing strategy for qPCR test that is still the most reliable SARS-CoV-2 testing strategy and still massively performed worldwide in addition to rapid tests. Also, to take into consideration is the fact that in developing countries, the access to antigen/rapid test is extremely limited and the vast majority of COVID-19 test are performed by PCR in certified laboratories and hospitals.
As we showed in the manuscript, we validated our method on 70 COVID-19 adult patients in an actual hospital setting and performed rigorous statistical analysis. Our results showed the feasibility of implementation and advantages of our method including cost reductions and minimizing biosafety hazards when collecting the samples. Moreover, we can argue that we routinely perform 30-50 tests each week, and we have not detected problems with the use of capillary tubes as patients are carefully instructed on how to use the provided saliva collection kit. Additionally, we compared hRP Ct values of saliva taken with microcapillary tubes vs. pippeted using a micropipette and there were no statistical differences between the protocols (Figure 2G).

Reviewer 2 Report
The Manuscript submitted by Bustos-Garcia and entitled “Development and testing of a low-cost inactivation buffer that allows direct SARS-CoV-2 detection in saliva” describes an evaluation of a inactivation buffer consisting of protease, detergent and RNase inhibitor. Various concentrations were evaluated (1:1, 2:1, 3:1) with a heat inactivation step. Inactivated specimens were then tested on singleplex and multiplex reactions using WHO primers and probes. Further work was done to optimize settings and look at stability of the buffer and stability of the specimens with the buffer. When optimized the Results of 70 previous positive samples were compared to the CDC assay using the QIAMP viral RNA extraction kit. Overall, the authors found around sensitivity of 85-87% depending on the probe. As expected, there was an increase in CTs compared to extraction with an average of 3.64 for N1 and 8.32 in N2. The data is well described, and sufficient controls were added to the study. However, most of the data is confirmatory to other extraction less methods used for saliva testing. Below are a few comments to be addressed.
Major comments
There have been some previous publications where the use of certain buffers added prior to heat inactivation can reduce amplification. Did you ever try to heat inactivate saliva and then add in the PKTP buffer for testing?
Minor comments
As it does look to have some decrease in stability, would it not be better to suggest making PKTP fresh, especially as a 24 hour at -20 does not add that much utility to a lab since it’s a short time frame. (ln273-276).
Author Response
Please see the attachment
We are grateful for the insightful comments from Reviewer 2, which have definitely improved our work.
To Reviewer 2:
- Major comments.
There have been some previous publications where the use of certain buffers added prior to heat inactivation can reduce amplification. Did you ever try to heat inactivate saliva and then add in the PKTP buffer for testing?
REPLY: We thank Reviewer 2 for her/his insightful comment. We are aware of previous works reporting that the sole inactivation of NSP was compatible with RT-qPCR; however, in our hands we had poor results by just heat-inactivating the saliva samples. Thus, we abandoned that strategy. We did not try adding the buffer after heat-inactivation and it would be very interesting to know whether the buffer improves sample stability and detection by acting as a protective agent after heat-inactivation or during this step. However, although we did not test this systematically, heat-inactivated saliva samples alone were detected only for samples with lower CT values compared with mixes of saliva:PBS and saliva:PK (PBS and proteinase K). We further proved that all the components that we were adding to the PK mixture enhanced RT-qPCR detection, especially for viral targets (see Figure 1).
- Minor comments
As it does look to have some decrease in stability, would it not be better to suggest making PKTP fresh, especially as a 24 hour at -20 does not add that much utility to a lab since it’s a short time frame. (ln 273-276).
REPLY: We thank Reviewer 2 for her/his comment. As Reviewer 2 insightfully notes, the buffer is more suitable when freshly prepared. It is even possible to prepare the buffer the night before the day of sample collection at schools, universities or working places. In such scenarios both, the buffer and the saliva samples mixed with the buffer were shown to be stable at RT until delivery to the lab for heat-inactivation and further processing (Figure 2). Following Reviewer 2’s excellent suggestion, we have added a statement suggesting the advantage of preparing PKTP fresh in the Results (lines 301-303).
